:ᐰ: PLOS | ONE

# Evolutionarily conserved susceptibility of the mitochondrial respiratory chain to SDHI pesticides and its consequence on the impact of SDHIs on human cultured cells

Paule Bénit[1], Agathe Kahn[1], Dominique Chretien[1], Sylvie Bortoli[2], Laurence Huc[3], Manuel Schiff(ORCID)[1,4], Anne-Paule Gimenez-Roqueplo[5,6], Judith Favier[6], Pierre Gressens[1,4], Malgorzata Rak[1], Pierre Rustin(ORCID)[1] *

**1** Université de Paris, NeuroDiderot, INSERM, Paris, France, **2** Université de Paris, INSERM, UMR-S 1124, Paris, France, **3** INRA UMR 1331 ToxAlim (Research Center in Food Toxicology), Université de Toulouse ENVT, INP, UPS, 180 Chemin de Tournefeuille, France, **4** Assistance Publique-Hôpitaux de Paris (AP-HP), Hôpital Robert Debré, Service de neurologie et maladies métaboliques, Paris, France, **5** Université de Paris, PARCC, INSERM, Equipe Labellisée par la Ligue contre le Cancer, Paris, France, **6** Assistance Publique-Hôpitaux de Paris (AP-HP), Hôpital Européen Georges Pompidou, Service de Génétique, Paris, France

* pierre.rustin@inserm.fr

## Abstract

Succinate dehydrogenase (SDH) inhibitors (SDHIs) are used worldwide to limit the proliferation of molds on plants and plant products. However, as SDH, also known as respiratory chain (RC) complex II, is a universal component of mitochondria from living organisms, highly conserved through evolution, the specificity of these inhibitors toward fungi warrants investigation. We first establish that the human, honeybee, earthworm and fungal SDHs are all sensitive to the eight SDHIs tested, albeit with varying $IC_{50}$ values, generally in the micromolar range. In addition to SDH, we observed that five of the SDHIs, mostly from the latest generation, inhibit the activity of RC complex III. Finally, we show that the provision of glucose *ad libitum* in the cell culture medium, while simultaneously providing sufficient ATP and reducing power for antioxidant enzymes through glycolysis, allows the growth of RC-deficient cells, fully masking the deleterious effect of SDHIs. As a result, when glutamine is the major carbon source, the presence of SDHIs leads to time-dependent cell death. This process is significantly accelerated in fibroblasts derived from patients with neurological or neurodegenerative diseases due to RC impairment (encephalopathy originating from a partial SDH defect) and/or hypersensitivity to oxidative insults (Friedreich ataxia, familial Alzheimer's disease).

## Introduction

Succinate dehydrogenase (SDH; EC 1.3.5.1), also known as electron transport chain (ETC) complex II (CII), is a universal and key component of the mitochondrial respiratory chain (RC) of all living organisms [1]. This protein transfers electrons derived from the oxidation of

**Data Availability Statement:** All relevant data are within the manuscript and its Supporting Information files.

**Funding:** No specific funding was initially provided for this work. After the work realization, PB and PR received a specific support grant from the French Association POLLINIS (https://www.pollinis.org/). General expenses were covered by grants from AAJI (Association pour l'Aide aux Jeunes Infirmes & aux Personnes Handicapées); AFAF (Association Française de l'Ataxie de Friedreich); Association OLY (Ouvrir Les Yeux); ANR MITOXDRUGS (ANR-16-CE18-0010). Sponsors or funders had no role in the study design, data collection and analysis, decision to publish, or preparation of the manuscript.

**Competing interests:** The authors have declared that no competing interests exist.

succinate to fumarate to a dedicated pool of ubiquinone without concomitant proton extrusion. The SDH complex is composed of four proteins (SDHA-D) that were largely conserved during evolution [2]. As part of the Krebs cycle, SDH is the only enzyme of the cycle that has no counterpart in any other cellular compartment and no soluble redox cofactors, such as $NAD^+$/NADH, which is possibly balanced out by the activity of other dehydrogenases; thus, SDH is irreplaceable in mitochondrial and cellular metabolism (S1 Fig). As a result, any blockade, even partial, of SDH activity can lead to changes in the cellular metabolome and have deleterious consequences for the cell [3]. Associated with its role in electron transfer, the enzyme is also favored in the RC for one-electron reactions through its iron-sulfur clusters, resulting in the generation of deleterious superoxides, especially in the presence of downstream SDH inhibitors (SDHIs) [4–6].

In humans, mutations of SDH-coding genes lead to pronounced defects of varying degrees in SDH activity that are associated with a wide spectrum of diseases, with frequent neurological expression [3]. Complete abolition of SDH activity results in profound modification of the metabolome, transcriptome and epigenome and to the development of various tumors and cancers [3, 7]. Generally, mitochondrial dysfunction is now recognized as a contributing factor to the early pathology of multiple human conditions, including major neurodegenerative diseases [8]. This finding is consistent with the role of mitochondria as signaling organelles with a multitude of functions, ranging from energy production as heat or ATP [9] to regulation of cellular metabolism [10, 11], energy homeostasis [12], stress response [13], and cell fate [14].

Given the central role of SDH, SDHIs are now widely used in agriculture worldwide to fight fungal proliferation [15]. These compounds are used on cereal crops; for preservation of fruits, vegetables, and seeds; and for care of public lawns and golf courses. However, because of the virtually universal function of SDH in cell respiration and mitochondrial metabolism, it can be assumed that any living organisms exposed to these substances could also be affected, and the consequences of exposure to SDHIs for non-target organisms appear a major issue. These effects depend on SDHI sensitivity of SDH, which may vary slightly based on the SDHI and the species used. The dose, time, frequency and duration of exposure; SDHI absorption and metabolization; and the eventual use of additional toxic substances are obvious factors to consider.

The fungicide qualifier associated with SDHIs infers some degree of specificity with respect to fungi, which cannot be confirmed based on basic scientific information comparing the action of the SDHI predecessor (carboxin) on SDHs from other species, including mammals [16]. This gap served as an incentive to investigate the effects of eight SDHIs, including the latest generation of molecules, on SDH activity and, more generally, on mitochondrial ETC quinone-dependent activities in four different species: *Botrytis cinerea*, as the target organism, and *Homo sapiens*, *Lumbricus terrestris*, and *Apis mellifera*, as non-target species. In addition to the predictable, general effect of the tested SDHIs against SDHs from different species, albeit with variable sensitivity, we observed that several of the latest-generation SDHIs also inhibited RC complex III (CIII) in addition to SDH (CII), indicating the potential binding of these SDHIs to other quinone-binding sites in the cell. We finally showed that in nonpermissive culture conditions for RC-deficient cells, these inhibitors readily impaired the growth and, ultimately, the survival of SDHI-treated human cultured cells presumably because of SDHI engendered oxidative stress.

## Material and methods

### Biological material

Four biological materials, *i.e.*, *B. cinerea* fungi, *A. mellifera*, *L. terrestris*, and human cultured cells were used to test the effect of SDHIs on the mitochondrial RC. To avoid interference with

contaminating factors and to maintain similar experimental conditions for the measurement of RC activities, different preparation methods had to be used for cultured cells, whole organisms and tissues [17].

### Botrytis cinerea

The mixed population of *B. cinerea* fungi used in this study was kindly provided by Dr. Joelle Dupont (Museum National d'Histoire Naturelle, Paris, France). Conidia from *Botrytis cinerea* were obtained from 15–30 d cultures grown on malt medium agar plates. Mycelial colonies were prepared by seeding conidia from 10 colonies in liquid medium (10 g of glucose, 2 g of $KH_2PO_4$, 0.5 g of $MgSO_47H_2O$, 1 g of $(NH_4)_2SO_4$, 2 g of yeast extract and deionized water up to 1 l, pH 6.2). Incubation was performed at 23°C until hyphae totally covered the surface of the bottle. The hyphae (9-cm-diameter disks) were harvested, washed twice with refrigerated distilled water and finally with mitochondria extraction medium containing 20 mM Tris (pH 7.2), 0.25 M sucrose, 40 mM KCl, 2 mM EGTA, 1 mg/ml BSA, and no cysteine. The hyphae were suspended in 150 ml of extraction medium, cut in small pieces (1 $cm^2$) and then disrupted at high speed for 1×2 sec and at low speed for 2×3 sec using a MultiMoulinex mixer (Moulinex, France). The suspension of disrupted hyphae was then filtered through two layers of nylon cloth (clearance 150 μm). The cloth was washed with 50 ml of extraction medium. After low-speed centrifugation (700 *g* for 15 min), the mitochondria were collected at 10,000 *g* for 20 min. The pellet was suspended in 250 μl of extraction medium, aliquoted (± 30 μl) and kept frozen at -80°C.

### Apis mellifera

*A. mellifera* individuals (n = 5) were a kind gift from Jacques Kemp (beekeeper; Saint-Rémy-lès-Chevreuse, Yvelines, France). Whole bodies of individuals frozen in dry ice, except heads, were homogenized using a 1 ml glass-glass Potter-Elvehjem homogenizer in an ice-cold medium consisting of 20 mM Tris (pH 7.2), 0.25 M sucrose, 40 mM KCl, 2 mM EGTA, and 1 mg/ml BSA. The homogenate was centrifuged at 1,500 *g* for 5 min. The mitochondria contained in the supernatant were spun down at 10,000 *g* for 10 min. The pellet of crude mitochondria was distributed as 20-μl aliquots and kept frozen at -80°C.

### Lumbricus terrestris

*L. terrestris* individuals (n = 5) were collected manually in a household compost in Epernon (France 28230) and identified by the person (Paule Bénit) in charge of this compost. Before use, the worms were washed 5 times with 1× phosphate-buffered saline. A fresh homogenate was prepared from the segments located between the prostomium and clitellum of the 5 individuals. Tissues were placed in 500 μl of an ice-cold medium consisting of 20 mM Tris (pH 7.2), 0.25 M sucrose, 40 mM KCl, 2 mM EGTA, and 1 mg/ml BSA and homogenized using a 1-ml glass-glass Potter-Elvehjem homogenizer. Low-speed centrifugation at 1,500 *g* for 5 min allowed heavy debris to be discarded. The supernatant was distributed as 50-μl aliquots and kept frozen at -80°C.

### Human cultured fibroblasts and HEK cells

Fibroblasts were derived from skin biopsies obtained from SDH-deficient patient, Friedreich ataxia (FRDA) patients, familial Alzheimer disease (FAD) and three anonymous healthy individuals, all of European ancestry. Written informed consent for research was obtained from patients and/or family members according to protocols in accordance to the Necker Hospital

(SDH-deficient patient; 1993) or Robert Debré (FRDA patient; 2011) Hospital ethical committees (Paris, France). None of these patients were recruited specifically for this study which used already available skin fibroblast cultures from the above mentioned Hospitals or from commercial source (FAD patient; Coriell Institute cell repository; AG08064). The SDH-deficient patient presented Leigh syndrome resulting from a homozygous R554W mutation in the SDH flavoprotein subunit (SDHA) [18]. The FRDA fibroblasts were from a female patient harboring a biallelic long GAA expansion (>2.6 kb) in the frataxin (FXN) gene (patient FRDA4; [19]. This expansion caused silencing of both the FXN and PIP5K1B genes [20]. Loss of pip5k1β function results in a decrease in PI(4,5)P2 levels, causing destabilization of the actin network and impaired Keap1-Nrf2 superoxide dismutase (SOD) signaling [19, 21]. As a result, these cells were highly sensitive to superoxides [21, 22]. The FAD fibroblasts were from a clinically affected male patient with FAD. Such Alzheimer's disease fibroblasts were previously shown to display abnormalities in the SOD-signaling-related Nrf2 pathway [23] and to be highly sensitive to oxidative insults [24, 25].

The cells were grown in T75 flasks using 10 ml Dulbecco's modified Eagle's minimal essential medium (DMEM) either containing 1 or 4.5 g/l glucose, 4 mM glutamine (glutamax; Gibco Thermo Fisher Scientific, MA), 2 mM pyruvate and 200 μM uridine (permissive medium, thereafter referred as GlucoMax) or lacking glucose, pyruvate and uridine but containing 4 mM glutamine (nonpermissive medium, thereafter referred as MitoMax; S1 Table). All media were supplemented with 10% fetal calf serum and 100 U/mL each penicillin and streptomycin. HEK293 human embryonic kidney cells were cultured in DMEM containing 4.5 g/L glucose, 4 mM glutamine, 10% fetal calf serum, 200 μM uridine, 2 mM pyruvate, and 100 U/mL each penicillin and streptomycin [9]. Cell pellets (1,500 $g$ for 5 min) were kept frozen (-80°C), and for biochemical studies, the pellets were permeabilized by two freeze-thaw cycles before enzyme assays [26].

To test SDHIs on growing skin fibroblasts, cells trypsinized from a confluent 75-cm$^2$ flask culture were seeded in 4×75-cm$^2$ flasks (20–25% confluency) containing GlucoMax medium (S1 Table). After allowing the culture to stand for one night, the medium was changed to the desired conditions, $i.e.$, GlucoMax medium or MitoMax medium (S1 Table), supplemented with the chosen amount of SDHI (control conditions compensated for DMSO used to solubilize SDHI; 14 mM max). The number of cells was estimated from random images using the ImageJ processing program for 15–20 days without changing the medium (one unique exposure to SDHI).

## Enzyme measurements and the potential effects of SDHIs

RC enzyme activities were spectrophotometrically measured using a pseudo-double-wavelength spectrophotometer (Cary 60, Agilent). Malonate-sensitive succinate quinone dichlorophenolindophenol (DCPIP) reductase (SQDR), as an indicator of the activity of the isolated SDH enzyme (isolated CII), and glycerol-3-phosphate quinone DCPIP reductase (GQDR), as a measure of the activity of the isolated glycerol-3-phosphate dehydrogenase (G3PDH), were assayed as previously described [17]. To study the effect of SDHIs on the RC, we had to take into consideration the fact that these inhibitors act through binding to quinone-binding sites in the RC. Accordingly, we observed that the extent of the inhibitory effects of these substances varies with the addition of exogenous quinones (Fig 1). When possible, the potential effects of SDHIs on RC enzymes were therefore studied under experimental conditions that did not require the addition of exogenous quinones. With this aim, we measured the malonate-sensitive succinate cytochrome c reductase (SDH+CIII), glycerol-3-phosphate cytochrome $c$ reductase (G3PDH+CIII) and antimycin-sensitive quinol cytochrome $c$ reductase (CIII) as

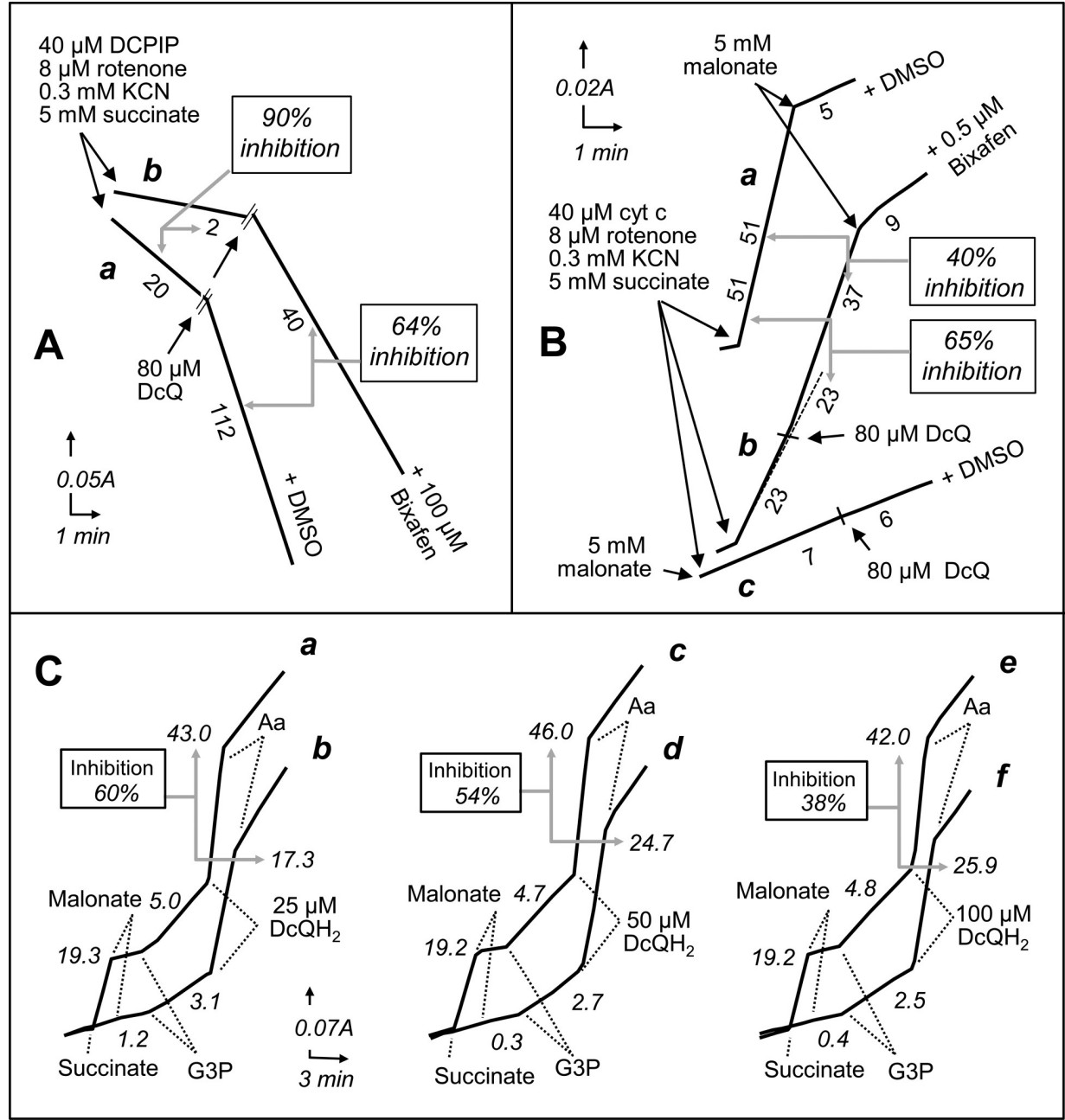

**Fig 1. Effect of SDHI (bixafen) on the respiratory chain as a function of quinone (decylubiquinone) concentration. A**, SDH activity from honeybee spectrophotometrically measured as reduction in DCPIP levels in the presence of succinate and exogenous quinone (SQDR). SDH activity was measured before and after the addition of exogenous decylubiquinone (80 μM) in the presence of bixafen (100 μM; trace b) or DMSO (trace a: control condition). Percent inhibition refers to the effect of bixafen in the absence or presence of decylubiquinone. **B**, Succinate cytochrome *c* reductase (SCCR) from freeze-thawed HEK cells measured in the presence of DMSO (trace a; control condition), bixafen (trace b), or malonate (trace c). Percent inhibition refers to the effect of bixafen (0.5 μM) measured before or after the addition of decylubiquinone (80 μM) on the malonate-sensitive cytochrome *c* reduction. **C**, Modulation by decylubiquinol of complex III inhibition by SDHI. A selected sequence of addition allows successful measurement of malonate-sensitive SCCR, glycerol-3-phosphate cytochrome *c* reductase (GCCR), and the antimycin-sensitive decylubiquinol cytochrome *c* reductase (QCCR) of HEK cells. The effect of bixafen (20 μM; traces b,d,f) on each of the three activities was observed by comparing the rates to the rates measured under control conditions (DMSO) and in the presence of various amounts of decylubiquinol: 25 μM (a,b), 50 μM (c,d), and 100 μM (e,f). Percent inhibition refers to the effect of bixafen on the antimycin-sensitive decylubiquinol cytochrome *c* reductase (CIII). Aa, antimycin; DCPIP (dichlorophenolindophenol); DcQ and DcQH2, the oxidized and reduced forms of decylubiquinone, respectively; G3P, glycerol-3-phosphate.

previously described [26]. The effects of eight different SDHIs were studied, namely, flutolanil, fluopyram, boscalid, fluxapyroxad, penflufen, penthiopyrad, isopyrazam and bixafen (S2 Fig), all of which are known to bind the quinone-binding site of SDH (S3 Fig). As DMSO, the solvent used for SDHIs and quinones, tends to reduce the activity of most RC enzymes, it was added in control conditions to compensate for DMSO added with SDHI (always below 25μl/ml, corresponding to 352 mM).

Total SOD activity was measured in freeze-thawed cell lysates resuspended in 50 mM $KH_2PO_4$ (pH 7.8). Activity was determined by monitoring the autoxidation of pyrogallol at 420 nm and expressed as IU/mg protein [27].

Protein concentration was estimated using the Bradford assay.

### SDHI $IC_{50}$ determination

The effects of eight different SDHIs (up to 500 μM; S2 Fig) were determined on four different biological materials. Because $IC_{50}$ are known to be affected by the actual absolute activity of the tested enzyme, these $IC_{50}$ were quantified using similar levels of SDH enzyme activity during the assay. $IC_{50}$ values were determined using the AAT Bioquest $IC_{50}$ calculator tool (https://www.aatbio.com/tools/ic50-calculator).

### Statistics

Data are presented as mean ± SD for all experiments. Statistical significance was calculated by standard unpaired t-test or one-way ANOVA with Bonferroni post-test correction for more than two conditions. A $p < 0.05$ was considered statistically significant (GraphPad Prism).

## Results

### Effects of SDHIs on the mitochondrial respiratory chain

We determined the effects of eight different SDHIs (S2 Fig) on the mitochondrial RC activity in freeze-thawed HEK cells, earthworms, honeybees, and *B. cinerea* fungi (Fig 2 and Table 1).

The $IC_{50}$ values for the different SDHIs with SDH were first estimated from the succinate cytochrome *c* reductase (SCCR) activity (Fig 2). All the SDHIs tested were found to exert an inhibitory effect on SDH regardless of the biological origin of the enzyme, albeit to varying degrees. As a reference, the ability of SDHIs to block SDH activity was studied on the enzyme of *B. cinerea*, the target organism, and was seen to vary depending on the SDHI, with flutolanil exhibiting relative ineffectiveness (Fig 2, Table 1). SDH showed highly pronounced sensitivity for the last-generation SDHIs, with an $IC_{50}$ below 0.1 μM for the fungal enzyme. Human SDH appears to be relatively less sensitive to fluopyram ($IC_{50}$ >150 μM) than to flutolanil, boscalid, penthiopyrad and fluxapyroxad ($IC_{50}$ values ranging from 18.6 to 2.1 μM). The earthworm enzyme appeared to be particularly sensitive to boscalid, flutolanil, and fluxapyroxad, whereas the honeybee enzyme was more sensitive to flutolanil and fluopyram than the enzymes of other non-target organisms. Last-generation SDHIs (S2 Fig) were particularly effective in blocking the human enzyme. The $IC_{50}$ values calculated for penflufen, isopyrazam and bixafen were all in the order of 1 μM or less. A similar variability among the effects of the various SDHIs was observed for the earthworm and the honeybee SDHs, which exhibited similar and higher sensitivity, respectively, to the last-generation SDHIs than to the SDHI predecessors. For most SDHIs, the $IC_{50}$ values were lower for the enzyme from the *B. cinerea* fungi, with the notable exception of flutolanil, which was particularly active against the earthworm and honeybee enzymes. Disturbingly, investigation of the SDHI sensitivity of SDH enzymes from only

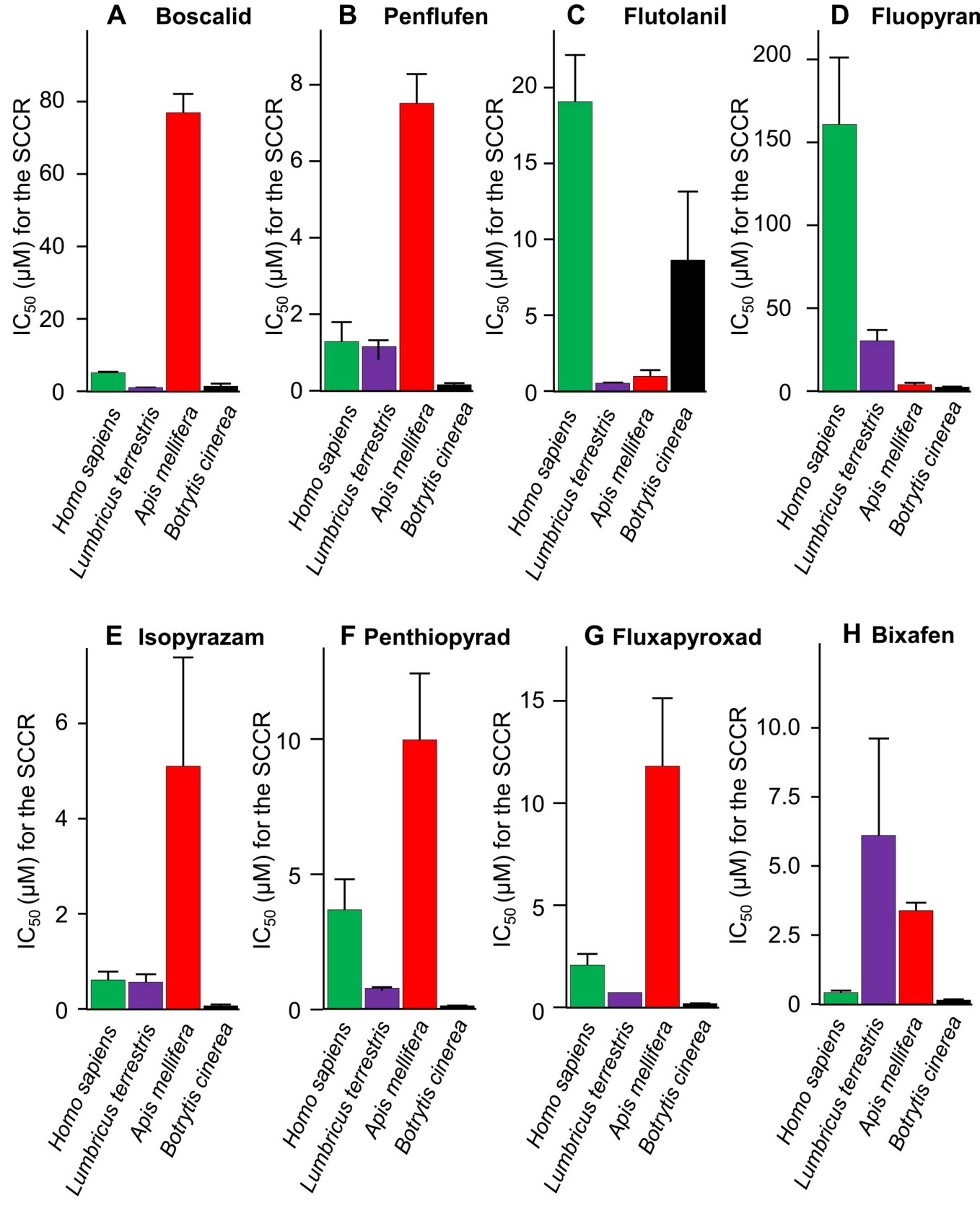

**Fig 2. IC$_{50}$ values of SDHIs on the succinate cytochrome $c$ reductase of *Homo sapiens*, *Lombricus terrestris*, *Apis mellifera*, and *Botrytis cinerea*.** A-H, IC$_{50}$ (n $\geq$ 3) of boscalid (A), penflufen (B), flutolanil (C), fluopyram (D), isopyrazam (E), penthiopyrad (F), fluxapyroxad (G), and bixafen (H) on the succinate cytochrome $c$ reductase of *Homo sapiens* (green), *Lumbricus terrestris* (purple), *Apis mellifera* (red), and *Botrytis cinerea* (black). Note the variable Y scales. SCCR, succinate cytochrome $c$ reductase.

4 species was enough to reveal that at least one of these species was quite sensitive to at least one of these SDHIs (Fig 2).

We next studied the effect of SDHIs on additional coenzyme Q-dependent RC activities, such as the activities of glycerol-3-phosphate cytochrome $c$ reductase (GCCR) and ubiquinol cytochrome $c$ reductase (QCCR), also known as RC CIII. Interestingly, isopyrazam and, even more efficiently, bixafen were found to also inhibit human GCCR activity (IC$_{50}$ 21.1 µM) (Table 1). Under our experimental conditions, no inhibitory effect of bixafen (up to 500 µM) could be detected on the glycerol 3-phosphate quinone-DCPIP reductase (not shown), while the molecule strongly inhibited CIII activity (IC$_{50}$ 15.5 µM), albeit less efficiently than it inhibited SCCR activity (IC$_{50}$ 0.34 µM) (Table 1). The effect of SDHIs was not restricted to human CIII and was exhibited by fluxapyroxad, penflufen, penthiopyrad, isopyrazam and bixafen on the fungal CIII and by penthiopyram, isopyrazam and bixafen on CIII from all 4 species. This unreported inhibitory effect of several SDHIs on RC CIII was modulated by the amount of exogenous quinones (DcQH$_2$) used to measure CIII activity, with 60 to 40% inhibition observed upon increasing DcQH$_2$ from 25 to 100 µM (Fig 1C). Notably, as adding reduced quinone, the substrate of the reaction, is necessary to measure CIII directly, the IC$_{50}$ values measured for CIII are only indicative. As expected, in case of CIII inhibition and similarly to the GCCR (Table 1), the NADH oxidase activity was also reduced (not shown), as would be observed following a treatment with rotenone or paraquat.

**Table 1. Comparative inhibitory effects (IC$_{50}$ values) of eight SDHIs on the RC activities of 4 different species.**

|  | *Homo sapiens* | *Lumbricus terrestrix.* | *Apis mellifera* | *Botrytis cinerea* |
|---|---|---|---|---|
| Flutolanil | SCCR: 18.7±3.5 µM<br>GCCR > 500 µM<br>QCCR > 500 µM | SCCR: 0.36±0.02 µM<br>GCCR: 450±70 µM<br>QCCR > 500 µM | SCCR: 0.8±0.5 µM<br>GCCR > 500 µM<br>QCCR > 500 µM | SCCR: 8.6±4.5 µM<br>GCCR > 500 µM<br>QCCR > 500 µM |
| Fluopyram | SCCR 160±40 µM<br>GCCR > 500 µM<br>QCCR > 500 µM | SCCR 30.6±7.6 µM<br>GCCR > 500 µM<br>QCCR > 500 µM | SCCR 3.8±0.3 µM<br>GCCR > 500 µM<br>QCCR > 500 µM | SCCR 0.2±0.1 µM<br>GCCR > 500 µM<br>QCCR > 500 µM |
| Boscalid | SCCR 4.8±0.2 µM<br>GCCR > 500 µM<br>QCCR > 500 µM | SCCR 0.5±0.3 µM<br>GCCR > 500 µM<br>QCCR > 500 µM | SCCR 76.7±6.0 µM<br>GCCR > 500 µM<br>QCCR > 500 µM | SCCR 0.8±1.1 µM<br>GCCR > 500 µM<br>QCCR > 500 µM |
| Fluxapyroxad | SCCR 2.1±0.7 µM<br>GCCR > 500 µM<br>QCCR > 500 µM | SCCR 0.71±0.07 µM<br>GCCR > 500 µM<br>QCCR > 500 µM | SCCR 11.7±3.4 µM<br>GCCR 205±34 µM<br>QCCR 300±5 µM | SCCR 0.095±0.008 µM<br>GCCR 60±56 µM<br>QCCR 67±53 µM |
| Penflufen | SCCR 1.3±0.3 µM<br>GCCR > 500 µM<br>QCCR > 500 µM | SCCR 1.13±0.16 µM<br>GCCR > 500 µM<br>QCCR > 500 µM | SCCR 7.5±0.7 µM<br>GCCR > 500 µM<br>QCCR > 500 µM | SCCR 0.13±0.06 µM<br>GCCR 80 ±57 µM<br>QCCR 433±58 µM |
| Penthiopyrad | SCCR 3.7±1.1 µM<br>GCCR 211±15 µM<br>QCCR 349±72 µM | SCCR 0.70±0.01 µM<br>GCCR 254±65 µM<br>QCCR 297±4 µM | SCCR 10±2 µM<br>GCCR 251±75 µM<br>QCCR 268±44 µM | SCCR 0.045±0.023 µM<br>GCCR 93.5±47 µM<br>QCCR 105±7.1 µM |
| Isopyrazam | SCCR 0.63±0.18 µM<br>GCCR 51.3±5.8 µM<br>QCCR 125±35 µM | SCCR 0.46±0.17 µM<br>GCCR 83±5 µM<br>QCCR 76±21 µM | SCCR 5.1±2.3 µM<br>GCCR 97±9 µM<br>QCCR 87±18 µM | SCCR 0.023±0.004 µM<br>GCCR 27.9±10.6 µM<br>QCCR 14.2±5.6 µM |
| Bixafen | SCCR 0.34±0.12 µM<br>GCCR 21.1±9.3 µM<br>QCCR 15.5±13.4 µM | SCCR 6.0±3.6 µM<br>GCCR 450±71 µM<br>QCCR $\geq$ 500 µM | SCCR 3.3±0.33 µM<br>GCCR 24.1±11.4 µM<br>QCCR 5.7±5.3 µM | SCCR 0.07±0.06 µM<br>GCCR 22.5±3.5 µM<br>QCCR 22.2±14.4 µM |

SCCR, succinate cytochrome $c$ reductase; GCCR, glycerol-3-phosphate cytochrome $c$ reductase; QCCR, quinol cytochrome $c$ reductase

## The SDHI-binding site is highly conserved during evolution

We next examined the molecular basis of susceptibility to account for the nonspecific effect of SDHIs on SDH from the different species studied. With this aim, we aligned the amino acid sequences of the three SDH subunits (B, C, D) involved in the SDHI- or quinone-binding site of the SDH (S3 Fig) by using the sequences available in NCBI from 22 organisms. Through examination of the SDH sequences of these 22 organisms, selected for the coverage that they provide in terms of evolution and/or susceptibility to SDHI exposure in nature, a fairly precise idea can be obtained for the probability of interaction of the SDHIs with SDH. We used the Cobalt alignment tool to provide an estimate of the conservation status of each amino acid [28]. A general examination revealed a high degree of conservation of a large portion of the SDH sequences throughout the 22 organisms regardless of the subunit considered (S4 Fig). This result is even more evident when one limits the comparison to the amino acids known to play an important role in the interaction of quinones with SDH or that have been shown to be the source of resistance to SDHIs when mutated [29] (S3 Fig). According to this analysis (Table 2), 100% identity among the 22 organisms was observed for more than half of the 15 amino acids considered, while all of these residues are annotated as conserved residues by the Cobalt tool [28].

## SDHIs and the viability of cultured human cells

Given the substantial effect of SDHIs on the activity of human SDH, we next assessed the effects of these compounds on the viability of human cells in culture (primary skin fibroblasts) (Figs 3, 4 and 5). It should be noted that the half-life of these extremely stable molecules

**Table 2. Amino acid residues involved in the ubiquinone-binding site and in SDHI binding in the SDH of 22 species (*Glomeromycota Rhizophagus irregularis, Botrytis cinerea, Rhizobium leguminosarum, Zymoseptoria tritici, Acropora digitifera, Medicago trunculata, Drosophila melanogaster, Nephila clavipes, Pieris rapae, Caenorhabditis elegans, Apis mellifera, Xenopus laevis, Danio rerio, Esox lucius, Sorex araneus, Oryctolagus cuniculus, Gallus gallus, Felis catus, Canis lupus, Ovis aries, Sus scrofa, Homo sapiens*).**

| | Involved in Q binding (% of the 22 cases) | Involved in SDHI binding (% of the 22 cases) |
|---|---|---|
| SDHB | **H267**: identity 100% | **P220**: identity 100% |
| | **I269**: identity 100% | **H267**: identity 100% |
| | **W224**: identity 100% | **W224**: identity 100% |
| | **W223**: identity 100% | |
| | **P220**: identity 100% | |
| SDHC | **S83**: identity 100% | L71: identity 48%; conserved 100% |
| | **R87**: identity 100% | W80: identity 19%; conserved 100% |
| | A84: identity 0.05%; conserved 100% | **S83**: identity 100% |
| | W80: identity 4; conserved 100% | A84*: identity 0.05%; conserved 100% |
| | P76: identity 9; conserved 100% | L85*: identity 1%; conserved 100% |
| | L71: identity 10; conserved 100% | **R87**: identity 100% |
| | | V88*: identity 0.05%; conserved 100% |
| SDHD | **D126**: identity 100% | A123*: identity 0.05%; conserved 100% |
| | **Y127**: identity 90%; conserved 100% | **Y127**: identity 90%; conserved 100% |

Residues are noted as being either identical across species or conserved according to the prediction from the *Cobalt* tool (set to 2 bits) included in the NCBI toolkit [28]. The SDH from *Botrytis cinerea* was used as the reference. Bold characters indicate residues known to be located in the SDH polar cavity

*amino acids found in the surface groove and not involved in carboxin binding.

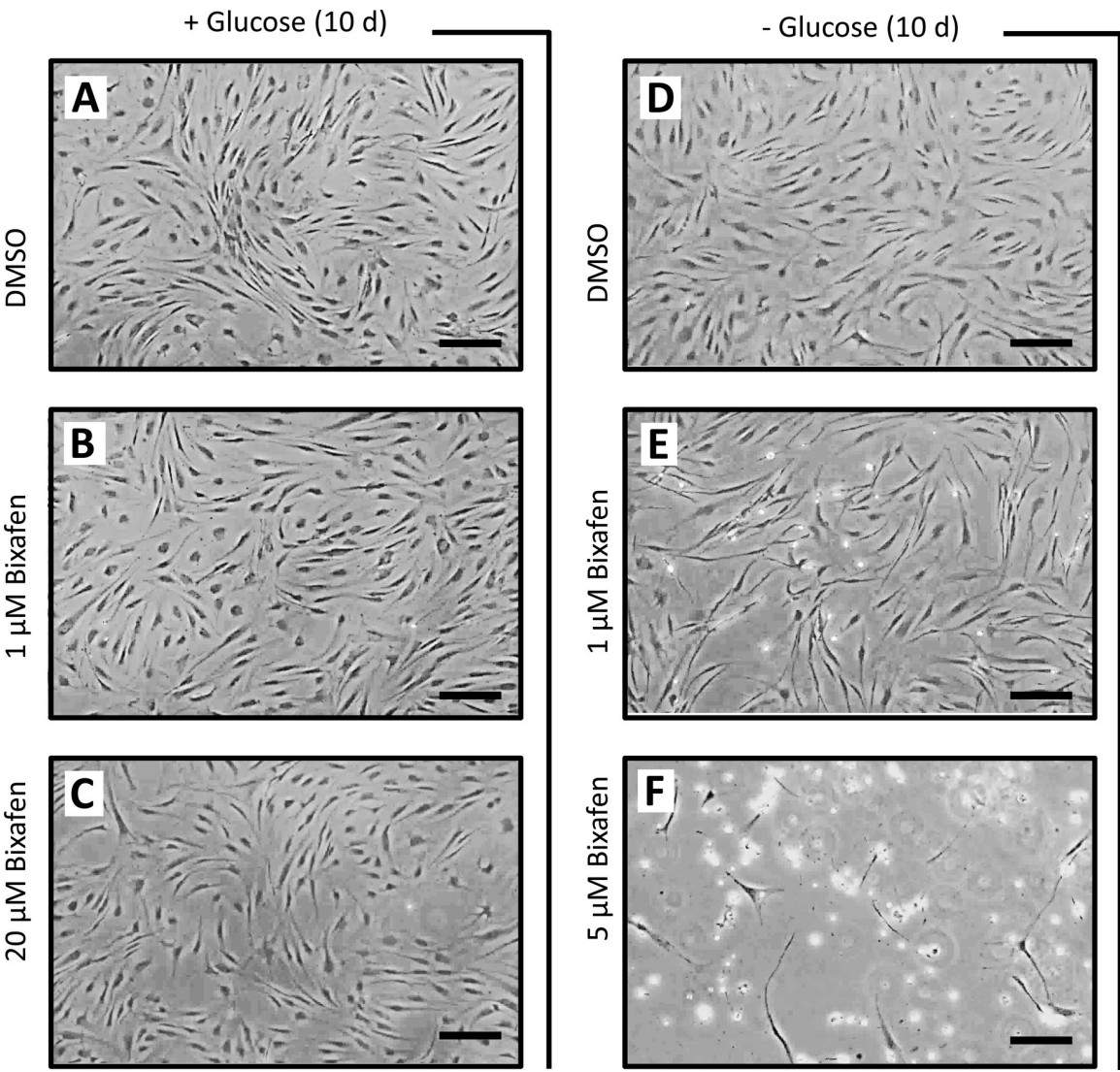

**Fig 3. Masked by glucose, the consequences of SDH inhibition by SDHIs in human cells in culture. A, D,** Control fibroblasts cultured for ten days in the presence (**A**) or absence (**D**) of glucose, glutamine being the sole source of carbon in the latter condition. The culture medium was not changed during the experiment. **B, E,** Cells were grown for a similar duration (ten days) without changing the medium in the presence of 1 μM bixafen. Note the presence of numerous white spots in the absence of glucose (**E**), indicative of the presence of dead cells in the low-density culture. **C, F,** Despite the presence of 20 μM bixafen, after 10 days of cultivation, no sign of cellular suffering was observed in the presence of glucose (**C**). A mirror experiment carried out in the absence of glucose (**F**) resulted in massive cell death at as low as 5 μM bixafen. In each figure, the black bar represents 200 μm.

exceeds months, thus reducing the number and frequency needed for their application on crops (1 or 2 *per* year according to distributors). We previously demonstrated using RC-defective fibroblasts from patients that the amount of glucose used under standard growing conditions (1 to 4.5 g/l) renders mitochondrial functions nonessential for long durations [30]. Accordingly, we observed that, despite the presence of 1 μM or even 20 μM bixafen, the cells grew actively for at least 10 days in a culture medium containing 1 g/l (5.5 mM) glucose (Fig 3A, 3B and 3C). Accelerated yellowing of the GlucoMax medium was nevertheless observed after 18 days in the presence of 20 μM bixafen, indicating increased acidification presumably due to abnormal accumulation of lactate [31]. Such lactic accumulation is one of the frequent

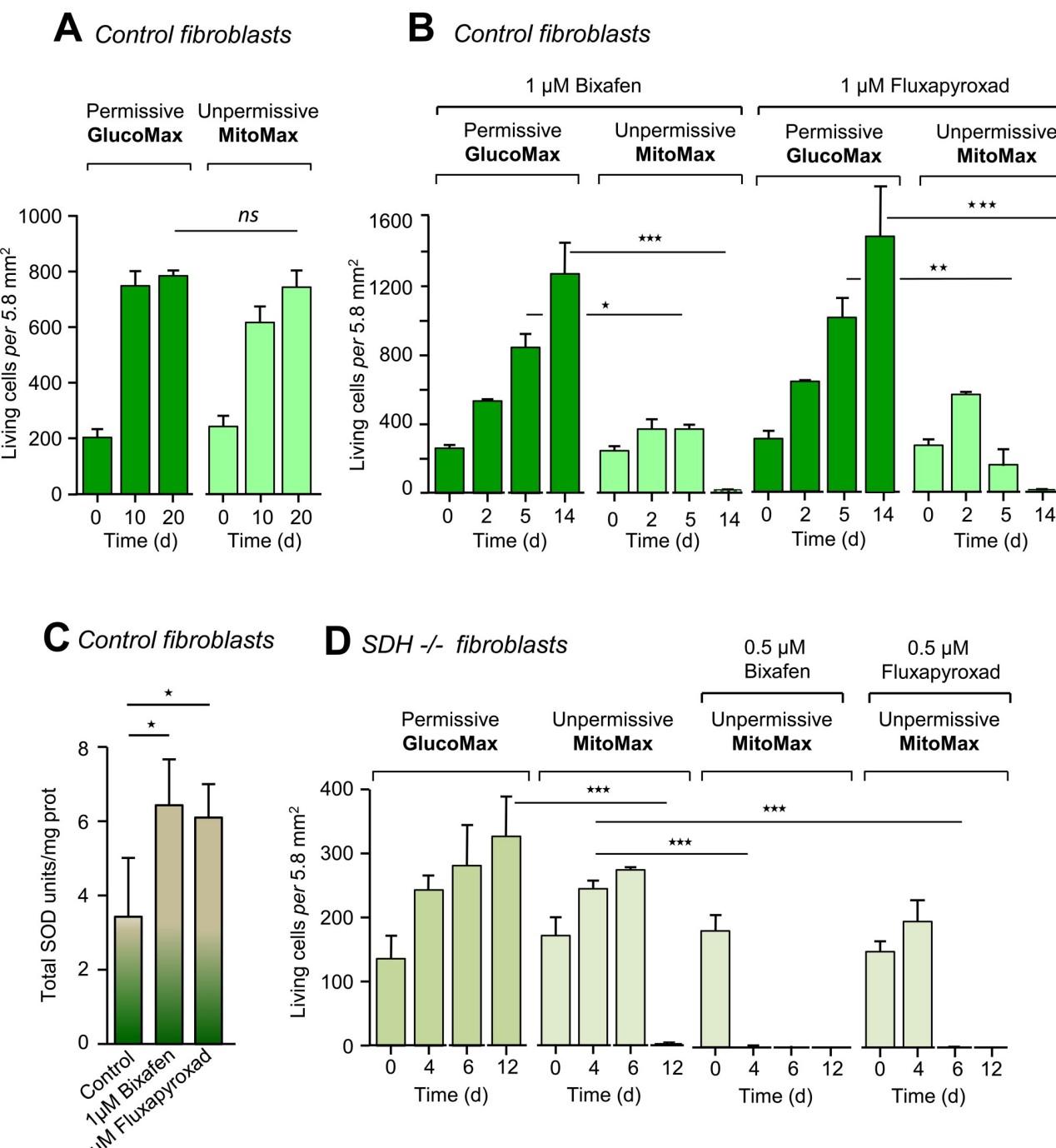

**Fig 4. Death of control and SDH-deficient patient cells induced by SDHIs in permissive GlucoMax and nonpermissive MitoMax media. A**, Control cells from three different individuals were allowed to grow in culture media either permissive or nonpermissive for defective mitochondria (see μ Table). In addition to glucose (4.5 g/l; hyperglycemic), the permissive medium contains 4 mM glutamine, 2 mM pyruvate (both a potential additional carbon source and a potent $H_2O_2$ scavenger), and 200 μM uridine, an amino acid, possibly in insufficient amount when a profound defect affects the terminal part of the respiratory chain (CIII or CIV; S1 Fig) [35]. **B**, SDHIs (1 μM) were tested (n = 3) in permissive and nonpermissive media. Both bixafen and fluxapyroxad resulted in massive cell death after 14 days of cultivation under nonpermissive conditions. Noticeably, similar attempts to test for the effect of boscalid at effective concentration (computed from $IC_{50}$ determination toward human SDH; Fig 2) resulted in cell death as well, but data were obscured due to the formation of a precipitate in the culture medium. **C**, Superoxide dismutase activity of control fibroblasts (n = 4) grown under permissive conditions (glucose 1 g/l) in the absence (control DMSO) or presence of 1 μM bixafen or 1 μM fluxapyroxad. **D**, Effect of bixafen and fluxapyroxad on patient fibroblasts with 60% residual SDH activity [18]. In nonpermissive medium, massive cell death was observed at 4 days of cultivation in the presence of 0.5 μM bixafen and 6 days for the same concentration of fluxapyroxad. Noticeably, the

used SDHI concentration (0.5 μM) is below the ADI for each of these molecules (0.58 μM) (S2 Fig). Culture media were not changed for the duration of the experiment (n = 3).

deleterious consequences observed in patients with mitochondrial RC defects and could lead to major brain dysfunction [32]. Notably, regular changing of the buffered cell culture medium artificially reduces the incidence of lactate accumulation, resulting from the effect of SDHIs on the mitochondria. In cells grown under this condition in the presence of 20 μM bixafen, the malonate-sensitive SCCR activity was found to be 80% inhibited, despite washing twice with 10 ml of PBS, trypsinization (2 ml), subsequent addition of 10 ml of trypsin stop buffer followed by cell centrifugation and a final wash in 50 ml of PBS. Simultaneously, we observed a significant doubling of total SOD activity after 13 days of cultivation in the presence of either 1 μM bixafen or 1 μM fluxapyroxad in a permissive medium (1 g/l glucose) (Fig 4C). Such induction of SOD activity is known to specifically reflect increased superoxide production by the mitochondrial RC [33], an increase that may account for the previously reported genotoxic effect of bixafen [34], and was previously observed in the presence of downstream inhibitors of the SDH complex [6].

Control fibroblasts were next cultured in a medium supplemented with glutamine (4 mM) as the only carbon source (Fig 3D–3F). In this challenging medium (nonpermissive MitoMax medium), energy metabolism became fully dependent on mitochondrial activity, allowing detection of the effect of SDH inhibition by bixafen. Under these conditions, using 1 μM bixafen for 10 days, we observed a significant reduction in the number of cells accompanied by the appearance of numerous dead cells (refractive white spots under the light microscope). A similar duration (10 days) of exposure to 5 μM bixafen resulted in the death of almost all cells (Fig 3F).

We next enriched the glucose-containing medium, making it even more favorable for the growth of RC-deficient cells, by increasing the glucose concentration to 4.5 g/l (25 mM) and adding 2 mM pyruvate and 200 μM uridine (permissive GlucoMax medium) (S1 Table). Upon cultivation in such a permissive medium for the growth of mitochondria-defective cells, fibroblasts can cope with most type of RC deficiency or inhibition, including the nonspecific effects of the SDHIs targeting CIII. Notably, the growth of the control fibroblasts was similar in both GlucoMax medium, which was permissive for defective mitochondria, and MitoMax medium, which lacked pyruvate and uridine and contained glutamine as the only carbon source (challenging medium for mitochondria) (Fig 4A). As expected, the presence of bixafen did not affect the growth of cells in the permissive GlucoMax medium. However, bixafen caused the death of almost all cells after 14 days of cultivation in the challenging MitoMax medium (Figs 4B and 5A). Notably, the observed effect was not restricted to bixafen, which was shown to inhibit both SDH (CII) and CIII (Table 1), and a similar effect was observed with fluxapyroxad, another new-generation SDHI (Fig 4B) that did not inhibit CIII in human cells (Table 1).

## SDHIs potently affect human cells with defective mitochondria

We next investigated the effect of SDHIs on cultured skin fibroblasts from patients with mitochondrial defects linked to either a partial SDH deficiency [18] or a hypersensitivity to oxidative insults due to impaired SOD signaling resulting from defective nuclear Nrf2 translocation [21, 23, 36]. The patient with an SDH deficiency presenting as Leigh syndrome [18] had 60% residual SCCR activity (1.8 ± 0.3 nmol/min/mg prot; control 3.0 ± 0.3 nmol/min/mg prot, n = 4) in fibroblasts. We first compared the growth of these SDH-deficient cells in the permissive GlucoMax medium to that in the challenging MitoMax medium (Fig 4D). After an initial

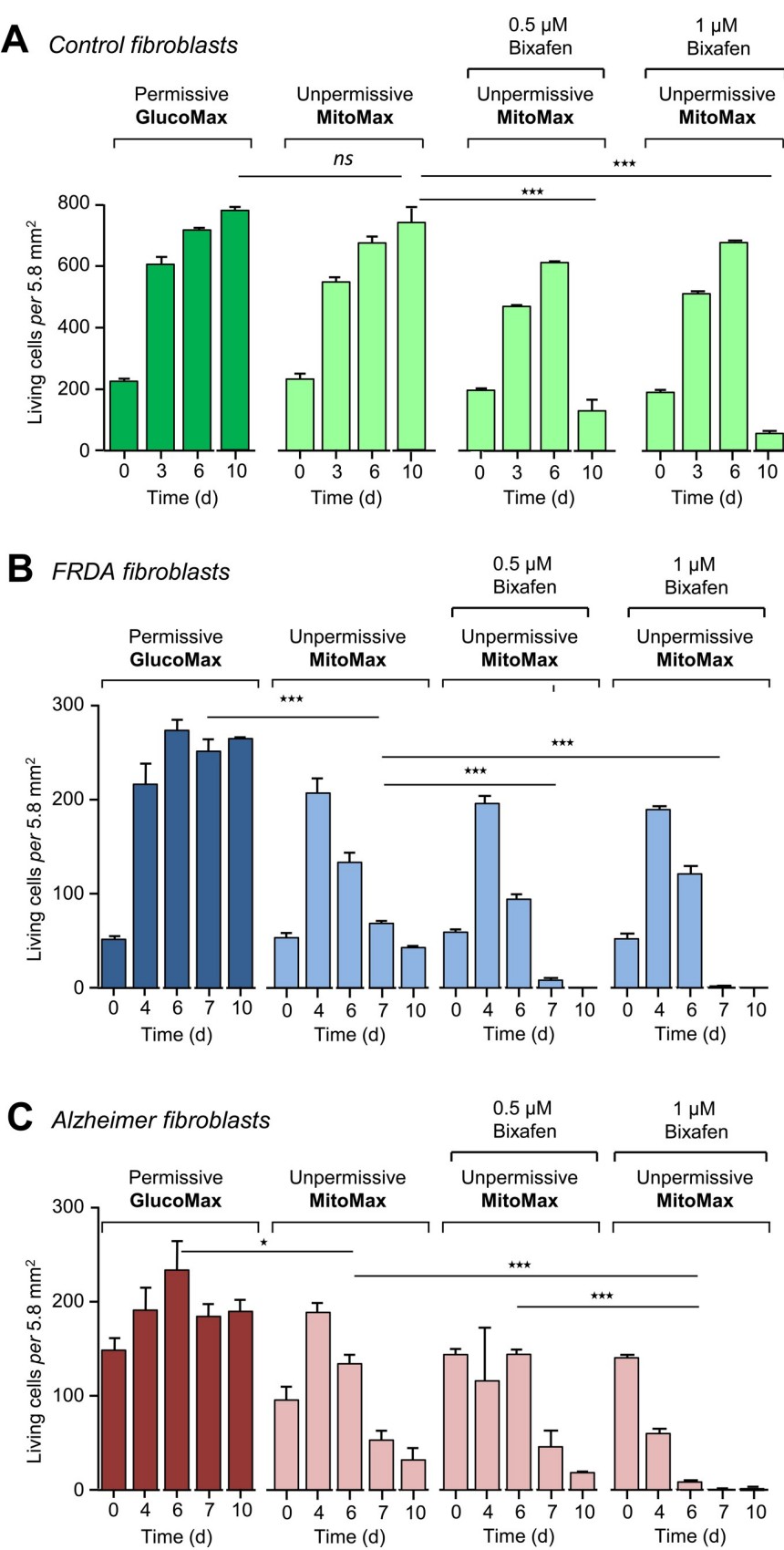

**Fig 5. Death of control, FRDA and FAD patient cells induced by bixafen. A**, Control cells were allowed to grow (n = 3) in either GlucoMax or MitoMax culture medium, which are permissive or nonpermissive for defective mitochondria (see Table 2), respectively, in the absence or presence of 0.5 µM or 1 µM bixafen. A similar experiment was carried out on fibroblasts from a Friedreich ataxia patient (FRDA fibroblasts) (**B**) and from a patient suffering from familial Alzheimer's Disease (**C**). Culture media were not changed for the duration of the experiment (n = 3).

slow growth rate observed in the MitoMax medium, cell death took over, resulting in near-total cell loss at 12 days. In the presence of 0.5 µM bixafen, total cell loss was observed after 4 days of cultivation (Fig 4D). In the presence of 0.5 µM fluxapyroxad, total cell loss was observed at day 6.

We next studied the effect of bixafen on fibroblasts selected for their hypersensitivity to oxidative insults from patients suffering from either FRDA [33] or FDA [24] [37] (Fig 5). Compared to control fibroblasts (Fig 5A), FRDA fibroblasts showed acceleration of cell death that was particularly marked at the highest concentration of bixafen (1 µM) for cells grown in MitoMax medium (Fig 5B). An identical, although less marked, phenomenon was observed for FAD cells (Fig 5C). After 4 days of growth in MitoMax medium, these cells exhibited susceptibility to 1 µM bixafen (Fig 5C), while control cells (Fig 5A) remained perfectly viable. Thus, abnormality in the functioning of mitochondria that directly affects the RC (defective SDH activity) or the ability to respond to oxidative stress (defective SOD induction) leads to hypersensitivity to the inhibitory effect of SDHIs.

## Discussion

The above data establish the inhibitory effects of eight SDHIs on SDHs from four biological species, namely, the fungus *B. cinerea*, the earthworm *L. terrestris*, the honeybee *A. mellifera*, and humans (*H. sapiens*), with varying inhibition strengths. This finding reflects the high conservation of the SDH B-D subunits that constitute the ubiquinone-reducing site of SDH (S2–S4 Figs). The lack of species specificity of the inhibitory effect of carboxin, the predecessor of most SDHIs, was recognized as early as 1976 [16], when the authors noted the extremely high affinity of carboxin for mammalian SDH [38]. Once widely sprayed as a fungicide on Canadian wheat, carboxin, used in doses similar to those used for the SDHIs derived from it, is indicated to be a major environmental hazard due to its nonspecificity [38]. In this work, we extended this observation to a series of SDHIs derived from carboxin, with which they share a common mode of action for inhibition of SDH.

The lack of species specificity of the inhibitory effect of SDHIs due to this mechanism of action is indeed known to agrochemical manufacturers, part of the SDHIs were reported to also be efficient against nematodes [39]. Similarly, the hazard represented by SDHIs for aquatic organisms has long been recognized, as indicated in the specification sheets, and confirmed by recent data obtained for *Danio rerio* with boscalid or penthiopyrad [40, 41] and for *Xenopus tropicalis* embryos with isopyrazam or bixafen [42]. Simultaneously, the toxic effects of boscalid on honeybees have been demonstrated to occur as long as the duration of exposure is sufficient [43]. The inhibitory effect of SDHIs presumably extends to most living organisms, modulated by the propensity of each organism to absorb, metabolize and/or excrete each of the SDHIs. Thus, the leaves of plants that exhibit poor absorption of SDHIs will be spared as long as the pesticide does not come into contact with the roots, which are likely to absorb the SDHI. Finally, the toxicological interactions resulting from the frequent use of a mixture of pesticides have been largely ignored, especially those associated with compounds such as CIII inhibitors (*e.g.*, strobilurin), which are often added to commercial preparations of SDHIs [42]. The toxic effects on the fauna of additional permeabilizing agents which are found in several SDHI mixtures have been similarly largely ignored.

It is extremely hazardous to compare the $IC_{50}$ values obtained *in vitro* under laboratory conditions with the concentrations of SDHI that might result from the application of these pesticides on crops. One can however reliably determine the concentration of SDHI at the outlet of the diffusion nozzles. In the case of bixafen, it is about 0.6 to 1.8 mM according to the recommendations distributed to farmers, resulting in a cloud of nebulization, corresponding to 75–125 g/hectare of bixafen. The final exposition depends on numerous factors hard to control, including the spreading conditions, the nature of the ground, its vegetation cover, etc. From a regulatory point of view, the ADI for example for boscalid and bixafen amounts to 0.04 and 0.02 mg / kg / day, respectively (see legend S2 Fig). For an average adult (60 kg / blood volume 5 l), this correspond to a blood concentration of approximately 1.40 and 0.58 μM, respectively.

To date, SDHIs have been reported to potently inhibit SDH activity, but the effect of SDHIs on additional sites, especially in the RC, has not been described. Here, we show that several of the latest-generation SDHIs inhibit CIII in addition to SDH, possibly increasing both their fungicidal activity and toxicity toward other organisms. To date, this additional effect has not been considered in the marketing of any of these molecules. Interestingly, however, SDHI-specific non-target site resistance has been recently described as being indicative of at least a second site of SDHI action [44].

The effects of RC inhibition on the growth of human cells in culture are greatly dependent on the balance between glycolysis and mitochondrial activity. Thus, so-called rho° cells, which lack mitochondrial DNA and therefore lack RC activity, are auxotrophic for uridine but can generate ATP from available glucose through glycolysis [45], producing a large amount of lactate (S1 Fig). Notably, complete disruption of the terminal part of the RC (CIII, CIV) results in the blockade of dihydroorotate dehydrogenase [46]. This latter enzyme normally produces the orotate required for pyrimidine synthesis, which is mandatory for cellular proliferation (S1 Fig), making uridine supplementation necessary to for strong growth of cells defective in CIII or CIV activity [46] or completely lacking mitochondrial DNA and therefore lacking a functional RC [45]. This is demonstrated by comparison of the permissive GlucoMax medium and the nonpermissive MitoMax medium, where the latter alone allows the detection of cells harboring mitochondria with deficient RCs. Notably, the use of this nonpermissive MitoMax medium is not intended to represent a given physiological situation but to allow detecting a possible deleterious effect of a given substance toward mitochondria. This MitoMax medium should be used a standard for testing the toxicity of any substance targeting mitochondria in cell culture in the context of hazard assessment in non-target organisms. In this study, we showed effective application of this strategy for SDHI-treated cells. Indeed, using standard glucose-containing media makes it impossible to detect even severe blockade of SDH in human cells. These experiments suggest the need for reevaluation of SDHI toxicity in human cultured cells using mitochondria-challenging MitoMax medium containing glutamine as the sole source of carbon. In any case, given the complexity of the phenomena possibly resulting from a mitochondrial stress (overproduction of superoxides, lactate accumulation, shortage of ATP, metabolic imbalance, etc.) it is central to respect a sufficient duration ($>$15 days) for these evaluations, based on the concentration of the inhibitor used.

Another aspect shown by this study is the increased sensitivity of cells with defective mitochondria to SDHIs. This finding holds true regardless of the nature of the mitochondrial defect, directly resulting from a partial deficiency of the RC or from hypersensitivity to oxidative stress due to poor induction of SOD. Isolated and partial SDH defects are rare human conditions that are generally associated with neurological or cardiac phenotypes [3]. However, in addition to the profound SDH defects evidenced in phaeochromocytomas and paragangliomas or gastrointestinal stromal tumor tissues of patients with germline mutations in the SDH

tumor suppressor genes [47–50], a partial SDH deficiency has been observed in a number of human diseases that affect one or several organs, *i.e.*, FRDA [51], Barth syndrome [52], various leukoencephalopathies [53] [54] [55], asthenozoospermia male infertility [56], myopathy [57], rare hemolytic uremic syndromes and rhabdomyolysis [58]. This feature defines a subpopulation that might be particularly at risk when in contact with SDHIs. Our results suggest that in this subpopulation and but also in populations, such as FRDA and Alzheimer's disease, known to present increased susceptibility to oxidative stress, a very common feature among human diseases, SDHIs could contribute to acceleration of disease progression.

## Conclusions

This work first establishes that, similar to their predecessor molecule carboxin [16], all tested SDHIs inhibit SDH in all the species considered, albeit with varying efficiency. Moreover, the tested next-generation SDHIs containing a methyl-pyrazol moiety (S2 Fig) also inhibit RC CIII. This lack of specificity represents a major problem considering the current widespread use of these SDHIs. While this lack of selectivity might be the source of the efficiency of last-generation SDHIs, it might also constitute an additional hazard for the exposed organisms. Our study next established that the standard conditions used to test the eventual toxicity of SDHIs (as well as of any other mitochondria-targeted pesticides) can mask potential toxic effect. Thus, we recommend regulatory testing for determination of molecular toxicity and the use of MitoMax growth medium in which glutamine is the sole source of carbon.

Finally, we show that a pre-existing mitochondrial defect, such as partial SDH dysfunction or hypersensitivity to oxidative insults (FRDA, FAD), increases the susceptibility to SDHIs, suggesting a particular risk for individuals with such a dysfunction.

## Supporting information

**S1 Table. Permissive *versus* nonpermissive culture media for mitochondrial respiratory chain-deficient human cells.**
(PDF)

**S1 Fig. Succinate dehydrogenase: A crossroad between the respiratory chain and intermediary metabolism.** Fum, fumarate; KG, α-ketoglutarate; Succ, succinate; TAC, tricarboxylic acid cycle; I-V, the various complexes of the respiratory chain (II, complex II or SDH).
(PDF)

**S2 Fig. SDHI molecular structure.** The yellow and blue parts for each molecule are reminiscent of the predecessor carboxin structure (top). The red dotted lines underline the methyl-pyrazol moiety present in the different SDHIs of the recent generation that also inhibit respiratory chain complex III. Boscalid (ADI: 0.04 mg/kg/d) was introduced in 2003 in the USA by BASF, fluopyram (ADI: 0.012 mg/kg/d) in 2010 in the USA by Bayer, flutolanil (0.09 mg/kg/d) in 1981 in the USA by Nichino America, penflufen (ADI: 0.04 mg/kg/d) in 2012 in the USA by Bayer, isopyrazam (ADI: 0.03 mg/kg/d) in 2010 in England by Syngenta, penthiopyrad (ADI: 0.1 mg/kg/d) in 2011 in the USA by Dupont-Fontelis, fluxapyroxad (ADI: 0.02 mg/kg/d)in 2011 in France by BASF, and bixafen (ADI: 0.02 mg/kg/d) in 2011 in England by Bayer. ADI; Acceptable Daily Intake according to European regulations.
(PDF)

**S3 Fig. Amino acids that are involved in the SDH quinone-binding site or mutation of which confers SDHI resistance. A**, Crystal structure of the quinone-binding site of the porcine (*Sus scrofa*) heart mitochondrial SDH (EC 1.3.5.1) drawn based on information from the Protein Data Bank archive. Amino acids are numbered corresponding to the porcine sequence

using Roman characters, while the italic characters outside the figure correspond to the numbering of the *Botrytis cinerea* sequence. **B**, Schematic depiction of the ubiquinone-binding site of SDH featuring some of the amino acids that have been said to favor fungal resistance to SDHIs (Sierotzki and Scalliet 2013). Encircled by the dotted line, a simplified representation of SDHIs shows in yellow the part of the molecule accommodated by the polar cavity of the ubiquinone-binding site of SDH; the part accommodated by the hydrophobic pocket is shown in blue.
(PDF)

**S4 Fig. Sequence analysis of the amino acid compositions of succinate dehydrogenase subunits B, C, and D. A,** List of the 22 organisms selected for alignment with the references of each protein sequence as deposited in the NCBI database. **B, C, D,** For each subunit, the graphic representation illustrates the conservation (indicated in red) of the subunit among the 22 species selected, followed by the alignment showing the amino acids present at the positions (gray background/dark arrows) supposedly involved in the resistance of fungi to SDHIs or/ and in the binding of coenzyme Q (S3 Fig) [29]. Alignment was performed using the COBALT multiple sequence alignment tool (https://www.ncbi.nlm.nih.gov/tools/cobalt/re_cobalt.cgi).
(PDF)

## Author Contributions

**Conceptualization:** Paule Bénit, Laurence Huc, Manuel Schiff, Anne-Paule Gimenez-Roqueplo, Judith Favier, Pierre Gressens, Pierre Rustin.

**Funding acquisition:** Paule Bénit.

**Investigation:** Paule Bénit, Agathe Kahn, Dominique Chretien, Malgorzata Rak, Pierre Rustin.

**Methodology:** Pierre Rustin.

**Supervision:** Pierre Gressens.

**Validation:** Anne-Paule Gimenez-Roqueplo.

**Writing – original draft:** Paule Bénit, Sylvie Bortoli, Laurence Huc, Manuel Schiff, Anne-Paule Gimenez-Roqueplo, Judith Favier, Malgorzata Rak, Pierre Rustin.

**Writing – review & editing:** Sylvie Bortoli, Laurence Huc, Manuel Schiff, Anne-Paule Gimenez-Roqueplo, Judith Favier, Pierre Gressens, Pierre Rustin.

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
