## [Decision Letter · Decision Letter 0]

27 Sep 2019

PONE-D-19-21805

Evolutionarily conserved susceptibility of the mitochondrial respiratory chain to SDHI pesticides and its consequence on the impact of SDHIs on human cultured cells

PLOS ONE

Dear Dr. Rustin,

Thank you for submitting your manuscript to PLOS ONE. After careful consideration, we feel that it has merit but does not fully meet PLOS ONE’s publication criteria as it currently stands. Therefore, we invite you to submit a revised version of the manuscript that addresses the points raised during the review process.

We would appreciate receiving your revised manuscript by Nov 09 2019 11:59PM. To enhance the reproducibility of your results, we recommend that if applicable you deposit your laboratory protocols in protocols.io, where a protocol can be assigned its own identifier (DOI) such that it can be cited independently in the future. For instructions see: http://journals.plos.org/plosone/s/submission-guidelines#loc-laboratory-protocols

We look forward to receiving your revised manuscript.

Kind regards,

Annalisa Pastore

Academic Editor

PLOS ONE

Journal Requirements:

Reviewers' comments:

Reviewer's Responses to Questions

**Comments to the Author**

1. Is the manuscript technically sound, and do the data support the conclusions?

Reviewer #1: Yes

2. Has the statistical analysis been performed appropriately and rigorously? 

Reviewer #1: Yes

3. Have the authors made all data underlying the findings in their manuscript fully available?

Reviewer #1: Yes

4. Is the manuscript presented in an intelligible fashion and written in standard English?

Reviewer #1: Yes

5. Review Comments to the Author

Reviewer #1: This manuscript describes the effects of several Succinate dehydrogenase inhibitors (SDHIs) on different organisms (Botrytis cinerea, Homo sapiens, Lumbricus sp. and Apis mellifera) and on cell samples derived from healthy individuals as well as cells derived from SDH-deficient, Friedreich's ataxia and Alzheimer patients. The manuscript is generally well-written and presents some interesting results that will be of interest to the field. Of particular interest: the masking effect of glucose when testing the effects of SDHIs and the possible effects of the inhibitors on the environment and non-target organisms.

However, there are some aspects of the study presentation that would make these findings more accessible. Notably:

Material and Methods:

- Human cultured fibroblasts and HEK cells -

The two types of media are described at the beginning of this paragraph but are not named immediately. The two names GlucoMax and MitoMax are shown only at the end of the paragraph. This is a bit confusing.

Results section:

- Effects of SDHIs on the mitochondrial respiratory chain -

It would help to report the SDHIs concentrations used in agriculture as a comparison for the IC50 results.

- SDHIs and the viability of cultured human cells -

Half life of SDHIs? The experiments last for 12/14 days without the medium getting changed. The differences seen between the compounds could be due to differences in their respective half lives.

Fig. 4, B: only the graph for bixafen in permissive medium is reported. Could you show the fluxapyroxad graph in permissive medium as well?

Minor corrections:

L206/207: this sentence is a bit confusing. Try rephrasing.

L253: the IC50 is reported only for bixafen and not for isopyrazam. Both of them are discussed in this sentence, therefore I would include both IC50 values.

L256: the IC50 reported for bixafen on CIII is 15.3 µM, the table says 15.5 µM

L290: add "C" inside the brackets (Fig 3A, B and C) not (Fig 3A, B)

L321 to 324: this sentence is not very clear. Try rephrasing.

L329: the "D" needs to be bold.

6. PLOS authors have the option to publish the peer review history of their article (what does this mean?). If published, this will include your full peer review and any attached files.

Reviewer #1: No

---

## [Author Response · Author response to Decision Letter 0]

4 Oct 2019

Responses to reviewer’s comments

We first like to thank the reviewer for his/her positive appreciation of our manuscript.

We carefully considered all the points and try to change the manuscript according the suggestions. 

1 -Material and Methods: - Human cultured fibroblasts and HEK cells -

The two types of media are described at the beginning of this paragraph but are not named immediately. The two names GlucoMax and MitoMax are shown only at the end of the paragraph. This is a bit confusing.

We have now modified the manuscript as to refer to the two culture media at the head of the section.

Results section: - Effects of SDHIs on the mitochondrial respiratory chain -

It would help to report the SDHIs concentrations used in agriculture as a comparison for the IC50 results.

We have on one side attempted to calculate what could be these concentrations and for this introduced a new paragraph in the last discussion. The difficulty lies in the fact that if it is relatively easy to know what is distributed in the field, the actual result for the exposed organisms in the field is obviously dependent on so many factors that such calculation is much more erratic. In addition, for the sake of comparison, we have now intraduced when possible the ADI for each SDHI (eg legend of S2 Figure). 

- SDHIs and the viability of cultured human cells. Half life of SDHIs? The experiments last for 12/14 days without the medium getting changed. The differences seen between the compounds could be due to differences in their respective half lives.

The half-lives of these molecule exceed several months in the nature, an observation that rules out that such parameter could represent a major issue in our study

Fig. 4, B: only the graph for bixafen in permissive medium is reported. Could you show the fluxapyroxad graph in permissive medium as well?

We have modified the figure to insert a new set of data showing the lack of effect of Fluxapyroxad in permissive medium

Minor corrections:

L206/207: this sentence is a bit confusing. Try rephrasing.

Done

L253: the IC50 is reported only for bixafen and not for isopyrazam. Both of them are discussed in this sentence, therefore I would include both IC50 values.

The IC50 was already reported for both molecules in Figure 2

L256: the IC50 reported for bixafen on CIII is 15.3 µM, the table says 15.5 µM

Corrected

L290: add "C" inside the brackets (Fig 3A, B and C) not (Fig 3A, B)

Done

L321 to 324: this sentence is not very clear. Try rephrasing.

Hopefully better, at least we tried

L329: the "D" needs to be bold.

Done

---

## [Editor Report · Decision Letter 1]

8 Oct 2019

Evolutionarily conserved susceptibility of the mitochondrial respiratory chain to SDHI pesticides and its consequence on the impact of SDHIs on human cultured cells

PONE-D-19-21805R1

Dear Dr. Rustin,

We are pleased to inform you that your manuscript has been judged scientifically suitable for publication and will be formally accepted for publication once it complies with all outstanding technical requirements.

With kind regards,

Annalisa Pastore

Academic Editor

PLOS ONE

1) In the Methods, please specify what type of informed consent you obtained (for instance, written or verbal, and if verbal, how it was documented and witnessed).

2) In your Methods section, please provide additional information about participant recruitment, including the recruitment date range (month and year) and a description of how participants were recruited.
---

## [Editor Report · Acceptance letter]

25 Oct 2019

PONE-D-19-21805R1 

Evolutionarily conserved susceptibility of the mitochondrial respiratory chain to SDHI pesticides and its consequence on the impact of SDHIs on human cultured cells 

Dear Dr. Rustin:

I am pleased to inform you that your manuscript has been deemed suitable for publication in PLOS ONE. Congratulations! Your manuscript is now with our production department. 

With kind regards,

on behalf of

Dr. Annalisa Pastore 

Academic Editor

PLOS ONE